# Use of Biomarkers in Ongoing Research Protocols on Alzheimer’s Disease

**DOI:** 10.3390/jpm10030068

**Published:** 2020-07-24

**Authors:** Marco Canevelli, Giulia Remoli, Ilaria Bacigalupo, Martina Valletta, Marco Toccaceli Blasi, Francesco Sciancalepore, Giuseppe Bruno, Matteo Cesari, Nicola Vanacore

**Affiliations:** 1Department of Human Neuroscience, Sapienza University, 00185 Rome, Italy; remoligiulia92@gmail.com (G.R.); m.valletta@live.it (M.V.); toccaceliblasi.1529394@studenti.uniroma1.it (M.T.B.); sciancalepore.1679543@studenti.uniroma1.it (F.S.); giuseppe.bruno@uniroma1.it (G.B.); 2National Center for Disease Prevention and Health Promotion, Italian National Institute of Health, 00161 Rome, Italy; ilaria.bacigalupo@iss.it (I.B.); nicola.vanacore@iss.it (N.V.); 3Department of Clinical Sciences and Community Health, University of Milan, 20122 Milan, Italy; macesari@gmail.com; 4Geriatric Unit, Fondazione IRCCS Ca’ Granda Ospedale Maggiore Policlinico, 20122 Milan, Italy

**Keywords:** Alzheimer’s disease, biomarkers, drug development, clinical trials, diagnostic research

## Abstract

The present study aimed to describe and discuss the state of the art of biomarker use in ongoing Alzheimer’s disease (AD) research. A review of 222 ongoing phase 1, 2, 3, and 4 protocols registered in the clinicaltrials.gov database was performed. All the trials (i) enrolling subjects with clinical disturbances and/or preclinical diagnoses falling within the AD continuum; and (ii) testing the efficacy and/or safety/tolerability of a therapeutic intervention, were analyzed. The use of biomarkers of amyloid deposition, tau pathology, and neurodegeneration among the eligibility criteria and/or study outcomes was assessed. Overall, 58.2% of ongoing interventional studies on AD adopt candidate biomarkers. They are mostly adopted by studies at the preliminary stages of the drug development process to explore the safety profile of novel therapies, and to provide evidence of target engagement and disease-modifying properties. The biologically supported selection of participants is mostly based on biomarkers of amyloid deposition, whereas the use of biomarkers as study outcomes mostly relies on markers of neurodegeneration. Biomarkers play an important role in the design and conduction of research protocols targeting AD. Nevertheless, their clinical validity, utility, and cost-effectiveness in the “real world” remain to be clarified.

## 1. Introduction

In the last few decades, the notion of Alzheimer’s disease (AD) has substantially changed. The enhanced understanding of the underlying neuropathophysiological mechanisms has supported a gradual shift from a clinical-pathological conception of AD [1] to a more biology-oriented framework [2,3].

The progressive evolution of the AD construct has been mainly sustained by the identification of key biomarkers of the AD neuropathological process [4]. According to the latest research definition, AD can be diagnosed in vivo, independently of cognitive manifestations, by the means of biomarkers reflecting β-amyloid deposition, tau pathology, and neurodegeneration [3]. Although their use is currently recommended only for research purposes [3,5], biomarkers are increasingly used in specialist clinical settings [6,7]. Nevertheless, some concerns and methodological shortcomings still affect the use of candidate AD biomarkers in routine clinical care. First, their clinical validity and utility have not yet been fully proven, and inconclusive evidence exists about their analytical validity [8]. Moreover, the lifetime risk of dementia among individuals with positive AD biomarkers varies considerably. In this regard, it is estimated that most persons affected by preclinical AD based on abnormal biomarker status will not develop dementia in their lifetimes [9]. Furthermore, they are either expensive, invasive, or both.

Biomarkers often play an important role in the design and conduction of research protocols targeting AD [10]. They may improve the selection of participants and render more biologically homogeneous the sampled populations. Moreover, their use may help demonstrating target engagement by tested intervention, providing evidence on disease modification, informing analytic stratification, and monitoring adverse effects [10]. At the same time, the sustainability of biomarker’s translation into care models and the generalizability (i.e., transferability) of interventions to the “real world” should always be taken into account [11].

In the present study, we describe and discuss the state of the art of biomarker use in AD research by reviewing the ongoing protocols registered in the clinicaltrials.gov database, updating and extending previous analyses on the topic [12]. In particular, we focus on the adoption of candidate biomarkers of amyloid deposition, tau pathology, and neurodegeneration as (i) eligibility criteria; and (ii) study outcomes. This analysis may inform how AD biomarkers are currently adopted in the drug development process, shedding a light on potential methodological, clinical, and ethical issues. Moreover, this approach may provide useful insights into a public health perspective considering that AD biomarkers and diagnostic procedures tested in research protocols will likely be increasingly translated in the daily practice to support clinical activities (e.g., risk prediction [13]) and regulatory decisions (e.g., drug accessibility and reimbursability [14]).

## 2. Materials and Methods

### 2.1. Data Source and Search Strategy

Clinicaltrials.gov was used as the reference source for the present study. Clinicaltrials.gov is an online database provided by the U.S. National Library of Medicine, which collects information from clinical studies that are conducted worldwide on a wide range of diseases and conditions.

The database was explored on 10 May 2020, by using the following terms and fields in the advanced search function: “Alzheimer” [CONDITION OR DISEASE] AND “interventional studies (clinical trials)” [STUDY TYPE] AND (“not yet recruiting” OR “recruiting” OR “enrolling by invitation” OR “active, not recruiting”) [STATUS: RECRUITMENT] AND (“phase 1” OR “phase 2” OR “phase 3” OR “phase 4”) [PHASE]. No restriction on age, sex, date, and location was applied.

Two authors (M.V. and G.R.) independently screened the resulting records and assessed their adherence to the following inclusion criteria:i)Enrolling subjects with clinical disturbances and/or preclinical diagnoses falling within the AD continuum (i.e., preclinical AD, subjective cognitive decline, mild cognitive impairment, prodromal AD, and AD dementia) [3];ii)Testing the efficacy and/or safety/tolerability of a therapeutic (both pharmacological and non-pharmacological) intervention.

Thus, trials enrolling participants with non-AD dementias or healthy volunteers, and primarily aiming at investigating diagnostic procedures (e.g., a novel neuroimaging technique) were excluded.

The flow chart presented in Figure 1 shows the process of protocols selection.

### 2.2. Data Extraction

Two reviewers (M.V. and G.R.) independently extracted the following data from the selected protocols: NCT number; phase; status; study design; expected end date; sponsor; target condition; intervention; mechanism of action; primary and secondary outcome measure (s); planned number of participants. The use of the following AD biomarkers among the eligibility criteria and/or study outcomes was also assessed according to the AT (N) biomarker grouping [3]:-A, amyloid deposition: (i) low cerebrospinal fluid (CSF) Aβ42 or Aβ42/Aβ40 ratio; and (ii) positive amyloid positron emission tomography (PET) scan;-T, tau pathology: (i) elevated CSF phospho-tau (*p*-tau); and (ii) positive tau PET scan;-N, neurodegeneration: (i) atrophy on anatomic magnetic resonance imaging (MRI); (ii) elevated CSF total tau (*t*-tau); and (iii) fluorodeoxyglucose (FDG) PET hypometabolism.

Disagreements in the selection process and/or extraction of data were solved by consensus or by involving two additional reviewers (M.C. and M.T.B.).

### 2.3. Data Analysis

Data were provided for two categories of protocols: (i) those using biomarkers in the selection of participants; and (ii) those using biomarkers as study outcomes. These categories were partially overlapping because some studies adopted biomarkers both as eligibility criteria and endpoints. Percentages and median values were calculated to summarize the abstracted categorical and continuous variables. Chi-square and median tests were used to compare the methodological characteristics of protocols adopting versus non-adopting biomarkers to ascertain eligibility and/or as endpoints. Statistical significance was set at *p* < 0.05.

## 3. Results

### 3.1. Search Results

A total of 222 protocols of phase 1, 2, 3, and 4 interventional studies were retrieved by the structured search on clinicaltrials.gov. Sixty-four of them were subsequently excluded because they did not fulfill the predefined set of inclusion criteria. Specifically, 39 studies were not testing therapeutic interventions, but were evaluated novel diagnostic procedures. Moreover, 25 trials were recruiting participants with clinical conditions not falling within the AD continuum (e.g., healthy volunteers, patients with other neurodegenerative dementias). Thus, 158 protocols were ultimately included in the analysis, as shown in Figure 1. A high agreement (>90%) was reported by the two reviewers involved in the selection process.

### 3.2. Characteristics of Protocols Adopting Biomarkers

Overall, 92 out of the 158 identified interventional studies (58.2%) adopted candidate AD biomarkers (Table 1). Compared with protocols not using biomarkers (*n* = 66), these studies were more frequently in the earlier phases of drug development (i.e., phase 1 and 2), more commonly aimed at evaluating the safety/tolerability of the tested interventions or their impact on AD underlying pathophysiological mechanisms (e.g., amyloid deposition, tau pathology, neuroinflammation), and almost exclusively focused on the assessment of pharmacological therapies (Table 1). On the contrary, studies not adopting biomarkers mostly adopted cognitive, functional, and neuropsychiatric measures as primary outcomes and were more commonly finalized at testing non-pharmacological interventions.

A total of 27,566 participants will tentatively be enrolled in the 92 protocols using biomarkers, with estimated sample sizes ranging between 12 and 2400 (median 120) subjects. These protocols are mostly conducted in the U.S. and funded by the biopharma industry. Their reported starting dates vary from November 2010 to January 2021, with completion dates indicated as ranging between December 2019 and August 2026. In the majority of studies, patients with overt AD dementia were indicated as eligible for participation, whereas nearly 20% of protocols restricted participation to subjects with preclinical or prodromal AD. Measures of clinical improvement were indicated as primary outcomes by 41.3% of studies with biomarkers.

Concerning how biomarkers are planned to be used, 62 studies adopt biomarkers to ascertain the eligibility of participants, 66 as study outcomes, whereas 36 both as inclusion criteria and endpoints.

The main characteristics of the ongoing protocols on AD using biomarkers are resumed in the Appendix A.

### 3.3. Use of Biomarkers in the Selection of Participants

A total of 62 phase 1 (*n* = 7), phase 1–2 (*n* = 11), phase 2 (*n* = 26), phase 2–3 (*n* = 3), phase 3 (*n* = 14), and phase 4 (*n* = 1) studies are currently using AD biomarkers in the selection of participants. Twenty-one studies are testing anti-amyloid therapies, 7 anti-tau compounds, and 34 novel treatments with other mechanisms of action.

The biomarker-based criteria that is most commonly adopted to determine the eligibility of participants in the selected protocols are a positive amyloid PET scan (72.6% of the protocols) and a low CSF Aβ42 or Aβ42/Aβ40 ratio (48.4% of the protocols) (Figure 2). These two criteria are interchangeably used in 21 studies (33.9%). No study requires the documentation of amyloid positivity at both the CSF and PET assessment. CSF *t*-tau and *p*-tau levels, tau-PET, volumetric MRI, and FDG PET are instead less commonly used. The biologically supported selection of participants is mostly based on biomarkers of amyloid deposition (87.1% of the cases) rather than on those reflective of neurodegeneration (21.0%) or tau pathology (9.7%). Few studies are using combinations of biomarkers, often simultaneously measuring those indicative of amyloid deposition and neuronal injury (11.3% of the protocols) (Figure 2). No statistically significant difference was found when comparing the biomarkers adopted by phase 1 and 2 studies versus phase 3 and 4 studies (data not shown).

Only 6 out of the 36 protocols adopting CSF measurements in the ascertainment of participants’ eligibility indicated the considered cut-points (Table 2). The majority of protocols performing amyloid PET in the screening phase does not provide sufficient information about the adopted radiotracer (e.g., florbetapir, florbetaben).

### 3.4. Use of Biomarkers as Study Outcomes

Overall, 66 phase 1 (*n* = 8), phase 1–2 (*n* = 9), phase 2 (*n* = 31), phase 2–3 (*n* = 7), phase 3 (*n* = 9), and phase 4 (*n* = 2) studies are using AD biomarkers as study outcomes. Eighteen of them are testing anti-amyloid therapies, 3 anti-tau molecules, and 45 novel compounds with different biological properties. Eight protocols indicate biomarkers as primary outcomes, 46 as secondary outcomes, whereas 12 both as primary and secondary outcomes.

Changes of MRI or FDG PET findings are the most commonly adopted biomarker-based primary endpoints, being selected by 5 out of the 20 studies using biomarkers as primary outcome measures (Figure 3). Among protocols using biomarkers as secondary outcomes, the most frequently adopted measures are brain MRI (56.9% of the studies), CSF *t*-tau (36.2%), Aβ42 or Aβ42/Aβ40 (34.5%), *p*-tau (32.8%), and amyloid PET (31.0%). The use of biomarkers as study outcomes mostly relies on markers of neurodegeneration (48.1% of the cases) rather than on those of amyloid (21.6%) and tau (21.6%) pathology (Figure 3). In 21 studies, an ATN combination of biomarkers is adopted. No statistically significant difference was observed when comparing the biomarkers used as outcomes by phase 1 and 2 studies versus phase 3 and 4 studies (data not shown).

For the studies with biomarkers as primary outcomes, information on the sample size calculation has never been provided.

### 3.5. Use of Biomarkers Both as Eligibility Criteria and Study Outcomes

A total of 36 phase 1 (*n* = 5), phase 1–2 (*n* = 8), phase 2 (*n* = 13), phase 2–3 (*n* = 2), and phase 3 (*n* = 8) studies are using AD biomarkers both in the selection of participants and as study outcomes. These studies mostly include biomarkers of amyloid deposition among the eligibility criteria (91.7% of studies) and biomarkers of amyloid pathology (75.0%) and/or neuronal injury (75.0%) as endpoints.

## 4. Conclusions

In the present study, we provided a snapshot of the current use of candidate biomarkers in AD research protocols based on the data registered on the clinicaltrials.gov database. Overall, more than half of ongoing interventional studies targeting the AD continuum are adopting measures of amyloid deposition and/or tau pathology and/or neuronal injury. Biomarkers are used both for the selection of participants and to ascertain the efficacy of the tested interventions. They are mostly adopted by studies at the preliminary stages of the drug development process to explore the safety profile of novel therapies, and to provide evidence of target engagement (e.g., receptor occupancy) and disease-modifying properties. On the contrary, their use in trials primarily looking at improving the clinical manifestations of the disease is still limited. Indeed, only nearly 40% of studies with biomarkers adopted cognitive, functional, neuropsychiatric, and other clinical measures as their primary outcomes.

Based on our findings, a relevant number of clinical trials are incorporating measures of A to refine the clinical diagnosis of AD in participating subjects with dementia. It is well-established that a sizeable proportion of individuals clinically diagnosed with AD dementia have amyloid biomarkers incompatible with the diagnosis [15]. Patients with negative amyloid status tend to exhibit a slower clinical progression over time as compared with patients with amyloid positivity at the CSF and/or PET [15]. Introducing these procedures as enrichment tools can, therefore, allow to homogenize study samples, reduce the heterogeneity in clinical outcomes, and increase the probability of detecting drug-placebo differences. The same considerations can be extended to participants with mild cognitive impairment (MCI) in whom the demonstration of underlying amyloid pathology reduces the chance of observing alternative clinical trajectories (e.g., reversion to normal cognition) that may bias the study findings and their interpretation [16].

Different reflections may instead be applied to the preclinical stages of AD. In fact, among cognitively intact individuals, the detection of isolated amyloid positivity (or “Alzheimer’s pathological change” [3]) is associated with a more considerable heterogeneity of clinical outcomes, thus complicating the assessment of clinical endpoints. Moreover, most of the subjects testing positive for A biomarkers will likely not exhibit a significant cognitive and functional worsening over time [9,17,18]. For instance, it has been calculated that the lifetime risks of developing AD dementia of a 60-year-old man and a 60-year-old woman with amyloidosis are only 23% and 31%, respectively [9]. Therefore, there is the risk of including in research protocols a disproportionate number of subjects who will never develop the target condition, thus only exposing them to the risk of serious adverse events with an unfavorable harm/benefit balance. For these reasons, it is fundamental to develop and implement appropriate and ethical strategies of risk disclosure to avoid the misinterpretation of testing results and the generation of adverse psychological reactions [19]. It is noteworthy that only a few studies combined biomarkers of A and T in the selection of participants. Including measures of tau pathology is instead required to determine if someone who is in the Alzheimer’s disease continuum has indeed AD [3], consequently making more uniform the study populations and risk profiles of participants in clinical trials. 

Other important considerations are raised concerning the use of biomarkers as screening tools in clinical trials [20]. In particular, the “real world” transferability of interventions emerging as effective in restricted samples of asymptomatic individuals with positive biomarkers will hardly be sustainable for our healthcare systems. Pathological AD markers are very common (i.e., 46%) in representative samples of older people [21], and it has been estimated that about 46.7 million Americans had preclinical AD in 2017 [22]. So, how can we translate these procedures on such a large scale to identify candidates for future disease-modifying treatments? Other controversies such as the standardization of CSF cut-points, the interchangeability of PET radiotracers and CSF assays, and the agreement between amyloid testing procedures remain to be addressed.

Candidate AD biomarkers are also frequently used as study outcomes. Specifically, measures of neurodegeneration are often adopted as downstream biomarkers to provide objective evidence that a drug ameliorates neurodegeneration. Indeed, the attenuation of the neurodegenerative process may be demonstrated (or at least suggested) by changes in the rates of glucose metabolism in cortical neurons (FDG PET), brain atrophy (volumetric MRI), and neuronal injury (CSF *t*-tau) [23]. N biomarkers may, therefore, support disease modification if a drug–placebo difference is detected and coupled with a similar clinical benefit [10,24,25]. It is noteworthy that, to date, no AD biomarker has yet qualified as a surrogate endpoint [26], which is a measurement that substitutes for clinical endpoints and is expected to predict clinical benefit [25]. Different to other areas of clinical neuroscience, such as multiple sclerosis, no AD biomarker has been consistently associated with the clinical trajectory of the disease, especially in its preclinical and prodromal stages [25]. Markers of amyloid and tau pathology as outcome measures are instead mostly useful to demonstrate drug engagement. Especially in the early phases of drug development, longitudinal CSF and PET measures of amyloid and tau pathology may confirm that a given compound affects the target brain protein [10]. For instance, several anti-Aβ monoclonal antibodies have shown a dose and time-dependent plaque reduction at the amyloid imaging, reflecting target engagement [27].

All these reflections on the use of biomarkers in AD research reinforce the need pursue their thorough clinical validation, as various methodological issues still limit their translation in the daily practice [8]. In particular, there is a lack of studies analyzing the distribution of biomarkers in healthy subjects and their interindividual variability according to major clinical and biological characteristics (e.g., age, sex, frailty) [28]. This information would be of great relevance in the promise of personalized medicine approaches. Moreover, no candidate AD biomarker has yet passed all the phases on which the architecture of diagnostic research is based [29]. 

The present study has some limitations to be mentioned and discussed. Our analysis cannot constitute an exhaustive overview on the topic because there are other registries for clinical trials on AD ongoing worldwide besides clinicaltrials.gov. Only a limited amount of information is available in the registered protocols, making it challenging to achieve an unbiased analysis of the methodological aspects of the studies. For example, in most of the selected protocols, brain MRI was simply mentioned among the participant selection procedures to exclude non-AD diagnoses. However, we cannot determine whether, in some cases, volumetric sequences were performed. This may have led to an underestimation of results concerning the use of MRI as a biomarker of neurodegeneration. Moreover, only the biomarkers incorporated in the AT (N) framework were considered. There is emerging evidence that other measures may represent promising biomarkers to be used in clinical trials [10,25].

In conclusion, biomarkers are largely used in ongoing clinical trials targeting the AD continuum. Their adoption may relevantly improve the selection of participants and the assessment of the efficacy and safety/tolerability of novel treatments. Nevertheless, their use in drug development must be accompanied by a demonstration of their clinical validity, utility, and cost-effectiveness in the “real world”. Otherwise, there is a risk of an experimenting model of care and access to treatments that are unsustainable for our healthcare systems.

## Figures and Tables

**Figure 1 jpm-10-00068-f001:**
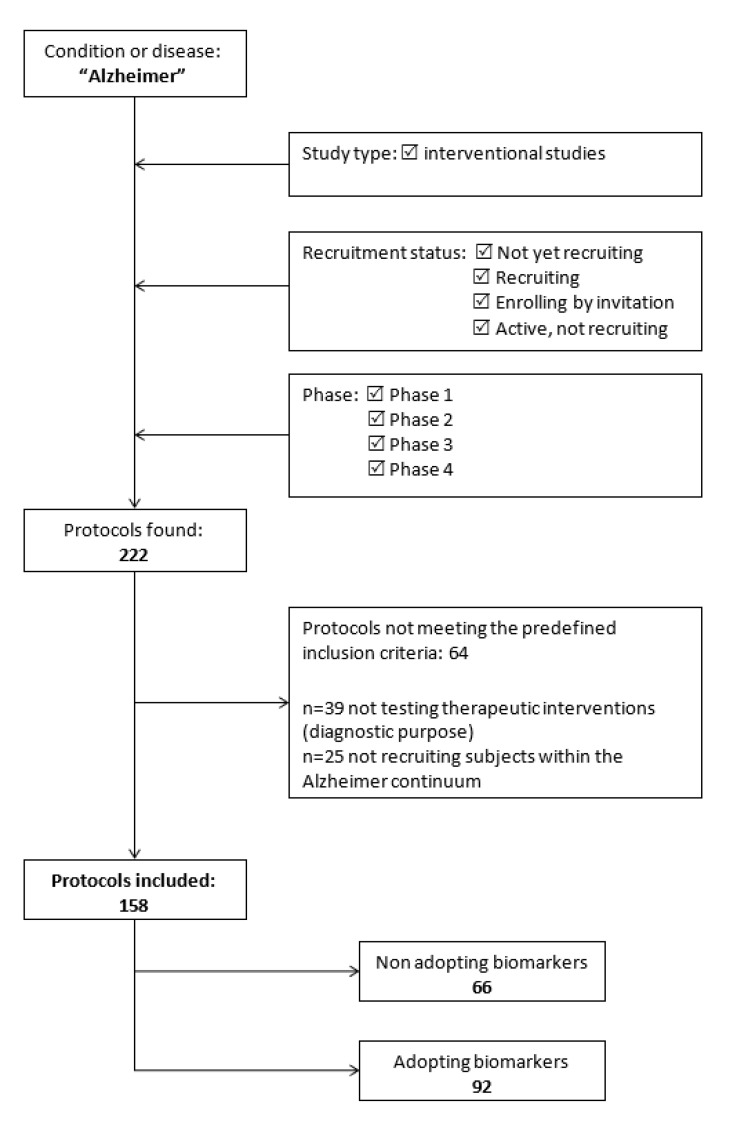
Flow-chart of protocols’ selection.

**Figure 2 jpm-10-00068-f002:**
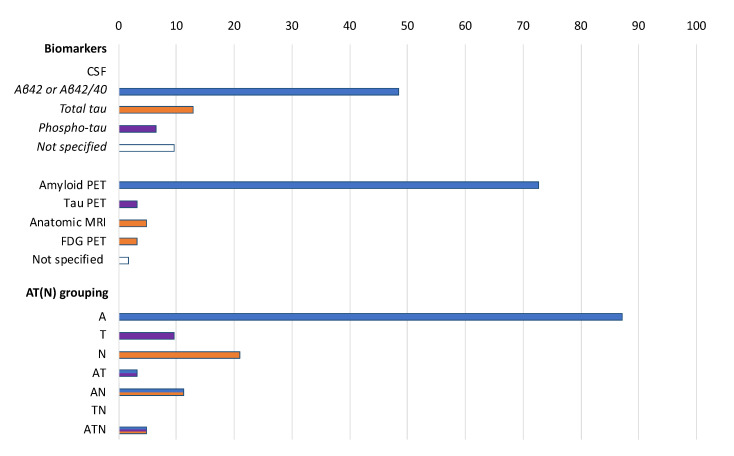
Biomarkers adopted to ascertain eligibility in the selected protocols (*n* = 62) according to the AT (N) grouping. Data are shown as %. A: aggregated Aβ or associated pathologic state; T: aggregated tau (neurofibrillary tangles) or associated pathologic state; N: neurodegeneration or neuronal injury. Colors refer to AT (N) biomarker grouping: blue: A; purple: T; orange: N.

**Figure 3 jpm-10-00068-f003:**
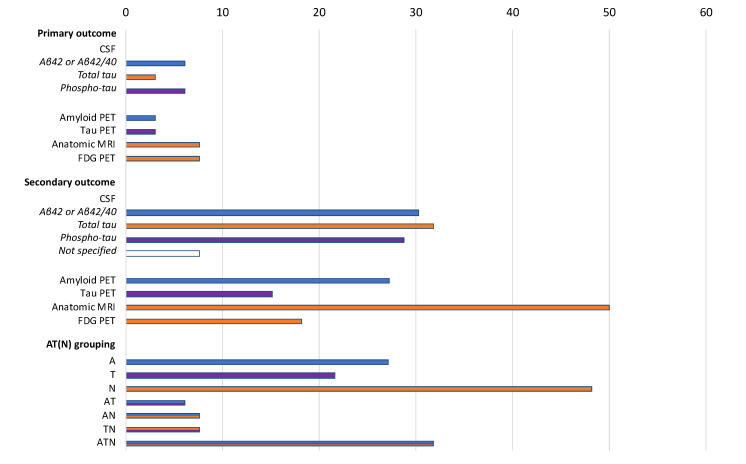
Biomarkers adopted as primary or secondary outcomes in the selected protocols (*n* = 66) according to the AT (N) grouping. Data are shown as %. A: aggregated Aβ or associated pathologic state; T: aggregated tau (neurofibrillary tangles) or associated pathologic state; N: neurodegeneration or neuronal injury. Colors refer to AT (N) biomarker grouping blue: A; purple: T; orange: N.

**Table 1 jpm-10-00068-t001:** Characteristics of the included protocols according to the adoption of biomarkers.

	Adopting Biomarkers (*n* = 92)	Non Adopting Biomarkers (*n* = 66)	*p*
Participants per study (*n*)	120 (43–382)	133 (43–267)	0.77
Phases			<0.01
Phase 1 and Phase 1–2	22 (23.9)	14 (21.2)	
Phase 2 and Phase 2–3	52 (56.5)	24 (36.4)	
Phase 3	15 (16.3)	18 (27.3)	
Phase 4	3 (3.3)	10 (15.2)	
Condition			0.21
Enrolling participants with AD dementia	74 (80.4)	58 (87.9)	
Not enrolling participants with AD dementia	18 (19.6)	8 (12.1)	
Main sponsor			0.12
Industry	56 (60.9)	32 (48.5)	
Other	36 (39.1)	34 (51.5)	
Primary outcome			<0.001
Safety	33 (35.9)	17 (25.8)	
Clinical improvement	38 (41.3)	47 (71.2)	
AD biological change	21 (14.6)	2 (3.0)	
Intervention			0.04
Pharmacological	87 (94.6)	56 (84.8)	
Non-pharmacological	5 (5.4)	10 (15.2)	

AD: Alzheimer’s disease; MCI: Mild cognitive impairment; SCD: Subjective cognitive decline. Data are shown as median (IQR) or *n* (%).

**Table 2 jpm-10-00068-t002:** CSF cut-offs adopted in the six protocols with available information.

	CSF Cut-Offs
NCT02547818	Aβ42 ≥ 180 pg/mL and ≤ 690 pg/mL
NCT02947893	Aβ42 < 600 ng/mL
NCT03061474	Aβ42 ≤ 600 pg/mL or *t*-tau/Aβ42 ratio ≥ 0.39
NCT03069014	Aβ42 < 550 ng/L or Aβ40/42 ratio < 0.89
NCT04079803	*t*-tau/Aβ42 ratio ≥ 0.28
NCT04191486	Aβ ≤ 1000 pg/mL and *p*-tau 181 ≥ 19 pg/mL

Values and measurement units are reported as specified in the protocols.

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
