# Peer review of "Use of Biomarkers in Ongoing Research Protocols on Alzheimer’s Disease"

_jpm, 2020, doi:10.3390/jpm10030068_

Round 1

Reviewer 1 Report

The authors present a comprehensive overview of the use of ATN biomarkers in current clinical trials. I have some minor points for improvement:

  • paragraph starting at line 145: I would start with the information in lines 151-159 and later mention the info in lines 147-150. Is it relevant to mention the funding of the studies? If so, it should also be mentioned for the studies not adopting biomarkers.
  • Lines 176-179: this implies that some protocols use both amyloid PET and CSF amyloid? could you indicate the percentage of protocols having both?
  • Studies that use biomarkers both as eligibility criteria and study outcomes (lines 213-218): are these also included in the two separate groups or is this a separate, third category. Please make this more clear in the introduction or methods.
  • line 226-227 (second part of sentence): I do not see this reflected in the results, thought you were looking at biomarkers for inclusion and as outcome measures. This seems related to the phase of drug development. 
  • lines 231-233: please describe this also in the results
  • lines 250-255: these statements are supported by one reference only, while many reports are published on the risk of developing dementia in SCD patients (=cognitively normal) with positive biomarkers. I suggest the authors nuance this statement by incorporating more evidence. 
  • Table 1: 'participants per study' instead of 'participants'?

Author Response

Point-by-point response to Reviewer

The authors present a comprehensive overview of the use of ATN biomarkers in current clinical trials. I have some minor points for improvement:

  • paragraph starting at line 145: I would start with the information in lines 151-159 and later mention the info in lines 147-150. Is it relevant to mention the funding of the studies? If so, it should also be mentioned for the studies not adopting biomarkers.

We have modified the paragraph according to the Reviewer’s suggestion.

  • Lines 176-179: this implies that some protocols use both amyloid PET and CSF amyloid? could you indicate the percentage of protocols having both?

Following the Reviewer’s comment, we have specified the percentage of studies adopting both CSF and PET amyloid assessment. We also clarified that no study required the documentation of amyloid positivity at both the CSF and PET evaluation:

“No study requires the documentation of amyloid positivity at both the CSF and PET assessment”.

  • Studies that use biomarkers both as eligibility criteria and study outcomes (lines 213-218): are these also included in the two separate groups or is this a separate, third category. Please make this more clear in the introduction or methods.

We completely agree with the Reviewer. We have now included the following sentences in the Methods section in order to clarify this point:

“Data were provided for two categories of protocols: i) those using biomarkers in the selection of participants; and ii) those using biomarkers as study outcomes. These categories were partially overlapping because some studies adopted biomarkers both as eligibility criteria and endpoints”.

  • line 226-227 (second part of sentence): I do not see this reflected in the results, thought you were looking at biomarkers for inclusion and as outcome measures. This seems related to the phase of drug development. 

We have modified the sentence as suggested by the Reviewer:

“Biomarkers are used both for the selection of participants and to ascertain the efficacy of the tested interventions”.

  • lines 231-233: please describe this also in the results

Following the Reviewer’s indication, we have included the following sentence in the Results section:

“Measures of clinical improvement were indicated as primary outcomes by 41.3% of studies with biomarkers”.

  • lines 250-255: these statements are supported by one reference only, while many reports are published on the risk of developing dementia in SCD patients (=cognitively normal) with positive biomarkers. I suggest the authors nuance this statement by incorporating more evidence. 

We completely agree with Reviewer. We have now included two additional references (#17 and #18) to better support that statements. 

  • Table 1: 'participants per study' instead of 'participants'?

We have modified the title of the table cell as suggested by the Reviewer.

Reviewer 2 Report

Title: Use of biomarkers in ongoing research protocols on Alzheimer’s disease

In this manuscript, the authors reviewed the current Alzheimer’s disease (AD) clinical trial registries. Specifically, the authors looked into the biomarkers used and assessed in each of the trials. This study provides an interesting and important overview of the CSF and imaging biomarkers used in the clinical trials, either as primary or secondary outcomes. Furthermore, the authors raised questions and provided insights in the discussion that is very helpful for digesting the information summarized. Overall, this is a good review. I would recommend it without further change.

Author Response

We thank the Reviewer for his/her positive comments.